# A Modelica Toolbox for the Simulation of Borehole Thermal Energy Storage Systems

**Julian Formhals** [1,2,*], **Hoofar Hemmatabady** [1,2], **Bastian Welsch** [1,2] , **Daniel Otto Schulte** [1] and **Ingo Sass** [1,2]

[1] Geothermal Science and Technology, Technical University of Darmstadt, Schnittspahnstraße 9, 64287 Darmstadt, Germany; hemmatabady@geo.tu-darmstadt.de (H.H.); welsch@geo.tu-darmstadt.de (B.W.); schulte@geo.tu-darmstadt.de (D.O.S.); sass@geo.tu-darmstadt.de (I.S.)

[2] Graduate School of Excellence Energy Science and Engineering, Technical University of Darmstadt, Otto-Berndt-Str. 3, 64287 Darmstadt, Germany

**\*** Correspondence: formhals@geo.tu-darmstadt.de; Tel.: +49-6151-16-20443

**Abstract:** Borehole thermal energy storage (BTES) systems facilitate the subsurface seasonal storage of thermal energy on district heating scales. These systems' performances are strongly dependent on operational conditions like temperature levels or hydraulic circuitry. Preliminary numerical system simulations improve comprehension of the storage performance and its interdependencies with other system components, but require both accurate and computationally efficient models. This study presents a toolbox for the simulation of borehole thermal energy storage systems in *Modelica*. The storage model is divided into a borehole heat exchanger (BHE), a local, and a global sub-model. For each sub-model, different modeling approaches can be deployed. To assess the overall performance of the model, two studies are carried out: One compares the model results to those of 3D finite element method (FEM) models to investigate the model's validity over a large range of parameters. In a second study, the accuracies of the implemented model variants are assessed by comparing their results to monitoring data from an existing BTES system. Both studies prove the validity of the modeling approaches under investigation. Although the differences in accuracy for the compared variants are small, the proper model choice can significantly reduce the computational effort.

**Keywords:** borehole thermal energy storage; Modelica; district heating; borehole heat exchanger; thermal resistance capacity model; model reduction

## 1. Introduction

Borehole thermal energy storage (BTES) systems are suitable for large-scale storage of thermal energy in the subsurface over periods of several months, thus facilitating seasonal storage of, e.g., solar thermal energy or waste heat [1–3]. The concept is principally based on storage of thermal energy in the subsurface, while the subsurface (i.e., soil or rock and pore water) serves as storage medium, the heat is injected and extracted with an array of borehole heat exchangers (BHE). For the construction of BHEs, a closed pipe loop is placed into a borehole and the remaining annular space between the piping and the borehole wall is backfilled to establish a good thermal contact between the pipes and the ground. Subsequently, heat can be injected or extracted by circulating a heat carrier fluid through the BHEs. The temperature difference between the heat carrier fluid and the rock determines the direction of heat transfer. For efficient operation of BTES systems, careful design based on a thorough understanding of its behavior is imperative. Operational conditions like the temperature and volume flow of the entering fluid have a strong impact on the storage performance. These conditions are defined by other components of the district heating system in which the BTES is embedded. Therefore,

a dynamic simulation of the whole system is necessary to account for all interactions and to achieve an accurate assessment of the heating system's performance. A language for the modeling and simulation of multi-domain physical systems is *Modelica* [4]. It uses an object-oriented equation-based approach facilitating state of the art modeling, simulation, and optimization of district heating systems [5–7]. While there are numerous analytical and numerical modeling approaches for the standalone assessment of BTES systems [8–10], only very few of those models are suited and implemented for dynamic system simulation software tools [11,12]. To overcome this and facilitate both accurate and efficient modeling and simulation of BTES systems in *Modelica*, a new model was developed. The model exploits typical characteristics of BTES systems for reducing the model complexity, while including the most important features of practical applications.

*1.1. State-of-the-Art Modeling Approaches for BTES Systems*

BTES systems are preferably built in areas with little to no groundwater flow to avoid larger dissipation of thermal energy by convective heat transport [13]. Thus, BTES models often disregard convective heat transport in the subsurface and focus on conductive processes [2,14]. However, the heat transport inside the BHE pipes is dominated by the circulation of the heat carrier fluid making convection the most important process. This difference in the prevailing heat transport processes inside and outside the BHE and their corresponding magnitudes have led to a variety of hybrid models [15–17]. These models usually consist of sub-models for the BHE and the surrounding ground. While the dimensions of the BHE models reach down to millimeters and seconds, the ground models focus primarily on scales from centimeters to dozens of meters and hours to years. This large range of magnitudes of the ground model has led to many approaches, which further divide the ground into a local and a global part. The local part covers the heat dissipation process around a single BHE and the global part considers only the heat transport between those local areas and the ground surrounding the storage [18]. Consequently, the division of the model decouples the different superposed heat diffusion processes of the storage system. When interactions with other model parts are disregarded, the underlying processes in all three models (the BHE model, the local, and the global model) exhibit distinct radial symmetric characteristics. This allows for a reduction of the model dimensions from 3D to 2D, which goes along with a significant reduction in the model's degrees of freedom (DoF). For each of the aforementioned sub-models, multiple analytical and numerical modeling approaches exist.

*1.2. Existing BTES Models for Dynamic System Simulation*

In general, there are many models for BHEs implemented into different software tools for dynamic system simulation. However, only a few take into account the thermal interactions between neighboring BHEs [16]. One of the first models—the Superposition Borehole Model (SBM)—was conceptualized by Eskilsson and Claesson [19]. It uses Thermal Resistance Models (TRM) for the representation of single BHEs. So-called "g-functions" are applied to factor in the interactions between neighboring BHEs in arrays with arbitrary geometric configurations. However, such functions have to be generated in advance by externally executed detailed numerical simulations. They solve the thermal response of a step-pulse for multiple line sources in a dimensionless form. Some methods have been developed to aggregate pulses further back in time, thus facilitating multi-year simulations with dynamic load conditions [20]. However, as transient storage operation conditions require sophisticated temporal superposition, the model is more suited for constant operation scenarios.

The most widespread BTES model is the duct ground storage model (DST) by Pahud and Hellström [11]. It reduces the global 3D thermal diffusion problem to a 2D problem assuming a radial symmetric storage geometry. As with the aforementioned SBM, the DST model also utilizes a TRM approach to represent single BHEs. The overall temperature inside the ground is obtained by the superposition of a local, a global, and a steady flux part. While the local and the global model are realized with finite difference models (FDM), the steady flux part is defined by analytical equations.

Both the SBM and the DST model have been implemented into *TRNSYS* [21], a software for dynamic simulations of thermal energy systems.

So far, the only BTES model that has been developed using the modeling language *Modelica*, is the Hybrid Step Response Model (HSRM) by Picard and Helsen [12]. For the representation of BHEs, the model uses a Thermal Resistance and Capacity Model (TRCM), as introduced by Bauer [22], increasing the short-time accuracy compared to TRM based models. Around each single BHE, a radial FDM is used for the local model. Moreover, the global temperature field is obtained using Javed's method [20], which is a simplified and more compact version of Claesson's approach. It analytically calculates the step response of multiple equal line sources. Temporal superposition of multiple pulses again renders the consideration of time-varying operation scenarios possible. As a result, the HSRM model facilitates accurate simulations of both short-term and long-term behavior of BTES systems with arbitrary designs. Nevertheless, the pre-calculation of the response functions for the global model are time-consuming. Consequently, the model is better suited for studies with a small number of different configurations. As a uniform temperature along all borehole walls is assumed, the applicability of the HSRM model is restricted to BTES systems with BHEs connected in parallel

Obviously, the existing models for simulation of BTES systems have certain drawbacks: Both the DST and SBM model disregard the thermal capacity of the backfilling inside the boreholes. Considering that the space between the pipes and the borehole wall—i.e., the backfilled space—yields the steepest thermal gradients, this imposes a serious limitation on the short-time accuracy of these models. Furthermore, there is no existing model that can simulate both the impact of different hydraulic circuitries as well as the resulting pressure losses. While parallel connection of BHEs is common for heat extraction boreholes, most BHE arrays for storage purposes show serial connection schemes. Therefore, disregarding serial connections as in the HSRM model poses a strong limitation for BTES applications. Moreover, the existing models assume homogeneous thermal properties inside the storage volume and do not allow for consideration of stratigraphic changes in these properties or of an upper BHE section with thermally insulating grout [23]. Additionally, there are some practical limitations of the models. For example, it is not possible to simulate two instances of the DST model simultaneously.

One general advantage of the modeling language *Modelica* is the possibility to create tool-independent models that can be used in different modeling and simulation environments. Even though *Modelica* standard conformity poses a challenging task for complex models, developers should try not to restrain the use of their model to a single software. Unfortunately, the HSRM does only work in the *Modelica* environment it was developed with.

To overcome these issues, a new open source *Modelica* toolbox for the Simulation of BTES systems, the *MoBTES* component library, has been developed. It is not restricted to a single simulation environment and has a modular structure, enabling modification of component type and modeling approaches. Important design features of BTES systems, like serial and parallel BHE connections, reversal of flow direction, pressure loss calculation, consideration of stratigraphic subsurface models, and partly insulated BHEs, are implemented.

## 2. Methods

### 2.1. MoBTES Modeling Approaches

To enable the efficient simulation of BTES systems in *Modelica* for both dynamic system simulations and large parameter studies, the component library *MoBTES* was created. It extensively utilizes object-orientation to allow for a high reusability of the BTES model components and an easy extensibility. It comprises components for the simulation of BHEs and BTES systems and uses components and interfaces from the *Modelica* standard libraries *Fluid Heat Flow* and *Heat Transfer* [4]. Generally, all models are built in accordance with the *Modelica Language Specification* version 3.4 [4], avoiding syntax and scripts specific to individual *Modelica* environments. Nevertheless, problems may occur since each environment handles the implementation of the *Modelica* language slightly different. The library

has been developed and optimized for user experience using *SimulationX* [24], but other software tools have been successfully tested as well. A short description of the library structure and the tested simulation environments can be found in Appendix A.

Analogous to the previously mentioned BTES modeling approaches, the model is subdivided into a global part, a local part, and a BHE part. In contrast to the actual shape of the storage under investigation, circular global, and local models are used, allowing for a reduction from 3D to 2D, by exploiting their symmetries. However, *MoBTES* offers the option to choose from circular, rectangular, and hexagonal layouts. The radii of the global and local models are calculated to result in volumes equivalent to those of the actual layout. The hypothesis that the size of the model volume is of higher importance than its shape is in line with Hellström [18] who concluded a detailed study on this matter.

Figure 1 shows the discretization of the modeled region into smaller volume elements that form the global model. The global model calculates heat flows on a large scale and therefor only considers the average temperature inside each volume element. Each global element, which is located inside the actual storage region, is connected to an element of the local model. This local element, in turn, is connected to a corresponding BHE element, such that it interconnects the BHE element with the global element. This setup of interfaces between the different sub-models links the borehole wall temperature and the average temperature inside the respective storage volume, i.e., the global element temperature. Exploiting the equation-based nature of *Modelica*, this allows for an independent mathematical description of the thermal processes within each sub-model.

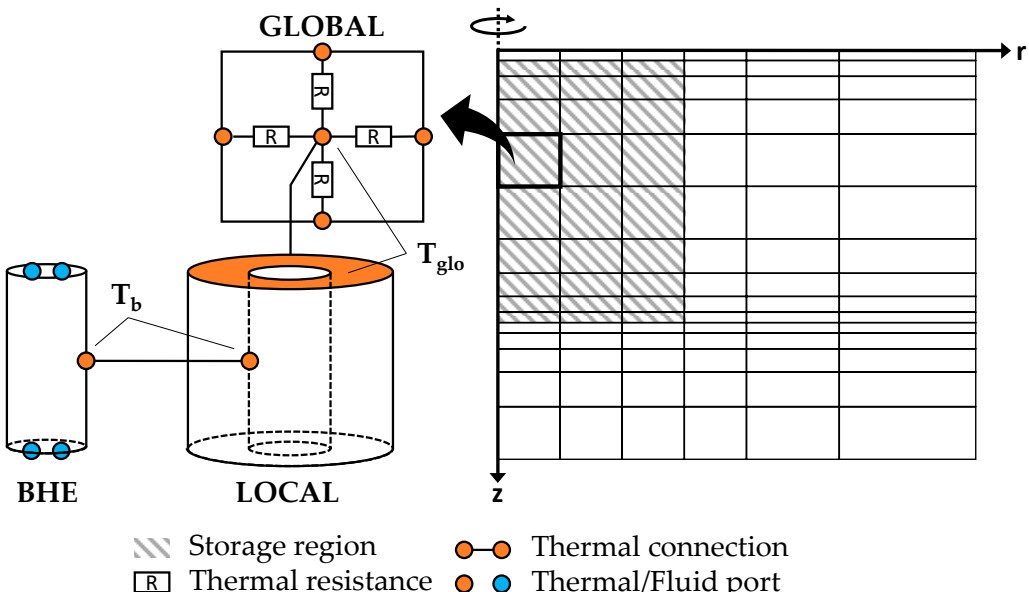

**Figure 1.** Mesh of a MoBTES model (**right**) and the fundamental connection scheme of the sub-models (**left**). The local element interconnects the global elements to their associated borehole heat exchanger (BHE) elements by giving a relation between the borehole wall temperature ($T_b$) and the mean volume temperature ($T_{glo}$).

The definition of interfaces between the sub-models together with *Modelica*'s object-oriented approach render the replacement of each sub-model possible and facilitate the utilization of different modeling approaches or component types. This enables the adaption of the model to the requirements of each application in terms of short- or long-term accuracy and computational effort. All currently available variants will be outlined in the following chapters.

### 2.1.1. Borehole Heat Exchanger Models

The BHEs are divided vertically into segments that are connected by flow ports. *MoBTES* offers two interchangeable approaches to solve the heat transport problem inside the BHEs, both of which

have their advantages and disadvantages. As the default approach, a TRCM after Bauer [22] is deployed, which takes account of the thermal capacities of the grout and therefore achieves a more accurate reproduction of the transient short-time behavior. The second approach disregards the grout capacities in a TRM. Consequently, the degrees of freedom (DoF) and thus the computing time are reduced. Both approaches contain models for Single-U, Double-U, and Coaxial BHEs. A detailed definition of the models is given by Bauer [22,25].

### 2.1.2. Local Heat Transport Models

The main purpose of a local model is to link the borehole wall temperature $T_b$ of the BHE segment it is connected with, to the temperature of its corresponding global volume element $T_{glo}$, which is defined as the average temperature inside this ground volume. There are currently two different approaches for the local model implemented in *MoBTES*: one generally more suited for transient operation scenarios and the other one rather for more steady conditions like step-response studies. For the more transient case, an FDM is used to represent the radial symmetric process around a single BHE segment. The approach, which is already known from the DST or HSRM models [11,12], divides the local volume into concentric ring elements (see Figure 2a). Following Eskilson and Claesson's [19] guidelines for radial meshing, the three innermost elements have an equal thickness and succeeding elements grow by a constant factor. As defined by Equation (1), the global temperature $T_{glo}$ is calculated by the weighted average ring temperatures. $C_i$ and $T_i$ are the thermal capacity and temperature of a single ring element, respectively, and $C_{loc}$ is the overall thermal capacity of the local volume.

$$T_{glo} = \sum_{i=1}^{n_{rings}} \frac{C_i}{C_{loc}} * T_i \tag{1}$$

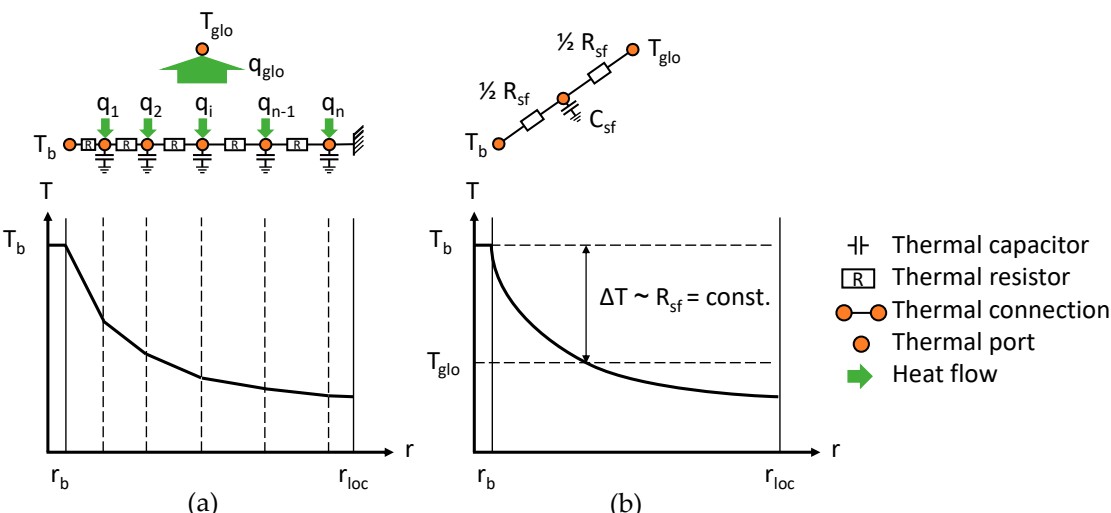

**Figure 2.** Local model concepts: (**a**) finite differences model and (**b**) steady flux model.

The heat flux $q_{glo}$ to or from a volume element is defined on the global level and distributed as $q_i$ among the n ring elements of the local model according to the weighting factors.

The second local model variant implemented in *MoBTES* is based on the steady flux part of the analytical solution for heat conduction inside a hollow cylinder with a fixed heat flow $Q_{sf}$ at the inner boundary and no flow of heat over the outer boundary [26] (see Figure 2b). This steady flux profile describes the temperature gradient within the hollow cylinder after initial transient processes have subsided. Its shape is time-independent and can therefore be used to calculate the temperature difference between the borehole wall temperature $T_b$ and the average volume temperature $T_{glo}$. The

relation between the steady flux $Q_{sf}$ and this temperature difference can be expressed as a resistance $R_{sf}$. The DST model uses this resistance to calculate heat transport processes for long time-scales [11].

Carslaw and Jaeger analytically investigated the conduction of heat in a hollow cylinder for different boundary conditions [27]. Their result for the above-mentioned case of a fixed heat flow $Q_{sf}$ over the inner boundary and no flow of heat over the outer boundary is given by Equation (2) and can be used to obtain a solution for the local volume temperature profile. It depends on time and radius and consists of three terms. The first one only depends on the energy injected over time and gives an expression for the average volume temperature $T_m$ (Equation (3)). The second term defines a time-independent radial temperature profile $T_o$, which corresponds to the profile shape under steady flux conditions, where $R_0$ is the thermal resistance between the borehole wall radius $r_b$ and radius $r$, $r_{loc}$ the radius of the local volume and $\lambda$ the thermal conductivity inside the volume (Equation (4)).

$$T(r,t) = Q_{sf}\left( \frac{1}{C_{loc}}t + R_0(r) - \sum_{i=1}^{\infty} R_i(r)\, e^{-\frac{t}{\tau_i}} \right) \tag{2}$$

$$T_m(t) = \frac{Q_{sf}}{C_{loc}}t = \frac{Q_{sf}}{c\,\rho\,\pi(r_{loc}{}^2 - r_b{}^2)}t \tag{3}$$

$$T_o(r) = Q_{sf}\, R_0(r) = \frac{Q_{sf}}{2\pi\,\lambda}\,\frac{r_{loc}{}^2}{r_{loc}{}^2 - r_b{}^2}\left( \ln\left(\frac{r_{loc}}{r}\right) - \frac{3}{4} + \frac{2r^2 - r_b{}^2}{4r_{loc}{}^2} + \frac{r_b{}^2}{r_{loc}{}^2 - r_b{}^2}\ln\left(\frac{r_{loc}}{r_b}\right) \right) \tag{4}$$

Finally, the third term describes the transition from the initially uniform temperature inside the hollow cylinder to the steady flux temperature profile. Its summands converge to zero with time. The speed of this process depends on the size of the respective time constants $\tau_i$ and is generally higher for summands with a higher index i. If this initial transition period is disregarded, Equations (5)–(8) can be used to obtain an expression for the steady flux resistance $R_{sf}$.

$$T(r_b, t) - T_m(t) = T_0(r_b) \tag{5}$$

$$= \frac{Q_{sf}}{2\pi\lambda}\left( \left(\frac{r_{loc}{}^2}{r_{loc}{}^2 - r_b{}^2}\right)^2 \ln\left(\frac{r_{loc}}{r_b}\right) - \frac{3r_{loc}{}^2 - r_b{}^2}{4(r_{loc}{}^2 - r_b{}^2)} \right) \tag{6}$$

$$= \frac{Q_{sf}}{2\pi\lambda}\left( \ln\left(\frac{r_{loc}}{r_b}\right) - \frac{3}{4} \right),\; r_{loc} \gg r_b \tag{7}$$

$$= Q_{sf}\, R_{sf} \tag{8}$$

Franke [28] introduced the idea of implementing a surrogate capacity $C_{sf}$ to this steady flux model to approximate the initial transient behavior until steady flux conditions prevail. Equation (9) gives the definition of this capacity, where $a$ is the thermal diffusivity of the local volume and $\tau_1$ the first and largest time constant of Equation (2). The resulting steady flux local model is depicted in Figure 2b. It was originally designed for dynamic optimization problems. Consequently, strong emphasis was put on computational speed.

$$C_{sf} = \frac{\tau_1}{R_{sf}} \simeq \frac{r_{loc}{}^2}{15\,a\,R_{sf}} \tag{9}$$

### 2.1.3. Global Heat Transport Model

Most BTES systems exhibit an axial symmetry. This symmetry is exploited by the global model to reduce the 3D problem to a 2D FDM model with axes in the radial and vertical directions. The subsurface domain is discretized into rectangular elements. Each of these elements represents the cross section of a ring of the modeled region. The average element temperatures, which are derived from the local model, serve as input for the calculation of the global thermal diffusion process. Furthermore, Dirichlet boundary conditions are used to define temperatures at the outer model edges. Thereby,

the temperature at the model's ground surface boundary is either set to the average annual ambient temperature or to a time-varying temperature defined by an input. To ensure a sufficient size of the modeled region, while simultaneously maintaining a low number of DoF, the size of the global elements outside of the storage region increases by a defined growth factor following the scheme of Eskilsson and Claesson [19].

### 2.2. Model Validation

The individual validities as well as the limitations of the sub-model approaches are demonstrated and discussed in detail in the original literature [18,25,28]. Therefore, the validation of the *MoBTES* model focuses on the functional interaction of the sub-models. To assess the accuracy of results, a parameter study was carried out for a large range of parameters, in which the energy balances of *MoBTES* models are compared to those of detailed FEM models.

In a case study, monitoring data from an existing BTES system is used to test *MoBTES'* ability to accurately reproduce real-world applications. The relative deviations $\delta Q$ of the charged and discharged energy as well as the resulting energy balance are calculated according to Equation (10).

$$\delta Q = \frac{Q_{\text{model}} - Q_{\text{reference}}}{Q_{\text{reference}}} * 100\% \tag{10}$$

#### 2.2.1. Parameter Study

To evaluate the long-term accuracy of the proposed model for both local model variants (FDM and steady flux), a parameter study is carried out, in which the average annual amounts of charged and discharged energy are compared to detailed 3D simulations. These benchmark simulations are performed in *FEFLOW* [29], a commercial finite element software tool for the simulation of groundwater flow, and mass and heat transport in porous and fractured media, which is frequently used for the simulation of BTES applications [8,30–32]. An operation period of ten years is simulated. In each year, the storage is charged for six months with a constant inlet temperature of 80 °C, and afterwards is discharged for another six months with an inlet temperature of 20 °C. All BHEs are connected in parallel and the volume flow rate is set to 2 l/s per BHE. The investigated parameter range is given in Table 1.

**Table 1.** Parameter range for the 3D finite element method (FEM) benchmark models.

| Parameter | Range |
| --- | --- |
| Number of BHEs | 4, 7, 9, 16, 19, 25, 36, 37, 49, 61, 62, 64, 81, 91, 93, 100, 121, 127, 130, 144, 169, 173, 196 |
| BHE length | 50 m, 100 m |
| BHE spacing | 3 m, 5 m |
| BTES layout | circular, rectangular, hexagonal |
| Local model variants | steady flux, FDM with 10 capacity nodes |

Closely-packed and symmetric BTES designs of rectangular, circular, and hexagonal shapes of up to 196 BHEs are investigated (Figure 3). For each of the resulting layouts, models with a minimal axial spacing of 3 m and 5 m between the BHEs as well as lengths of 50 m and 100 m are created. Finally, a total of 99 corresponding benchmark simulations are carried out. Each of those benchmarks is compared to two *MoBTES* simulations, one using the steady flux local model, and the other using the FDM local model with 10 capacity nodes. Moreover, all models in this study utilize the TRM variant for Double-U BHEs, since FEFLOW does not provide a TRCM model. A full list of all benchmark and *MoBTES* models, their parametrization, and aggregated results can be found in the supplementary data (Table S1).

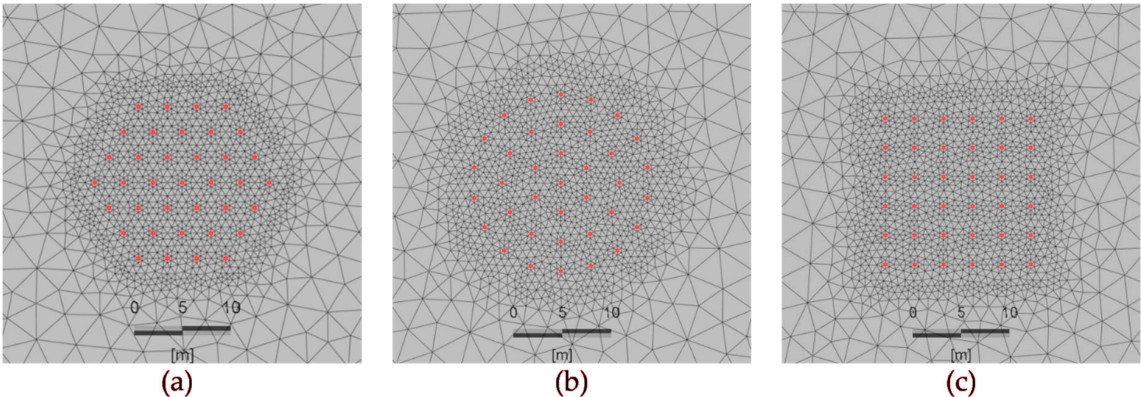

**Figure 3.** Top view on *FEFLOW* benchmark models with red markers for BHE positions: (**a**) hexagonal storage with 37 BHEs; (**b**) circular storage with 37 BHEs; (**c**) rectangular storage with 36 BHEs.

2.2.2. Case Study

To assess the predictive abilities of *MoBTES* concerning real storage operation, a BTES system installed in the Brædstrup solar district heating system (Denmark) [33] is simulated and the results are compared to monitoring data. The Brædstrup system consists of 48 BHEs with a length of 45 m each. Six BHEs are connected in series, resulting in eight strings of BHEs in parallel (see Figure 4). During charging operation, the flow through the BTES system is directed from the central BHEs to the peripheral BHEs. During discharging, the flow direction is reversed.

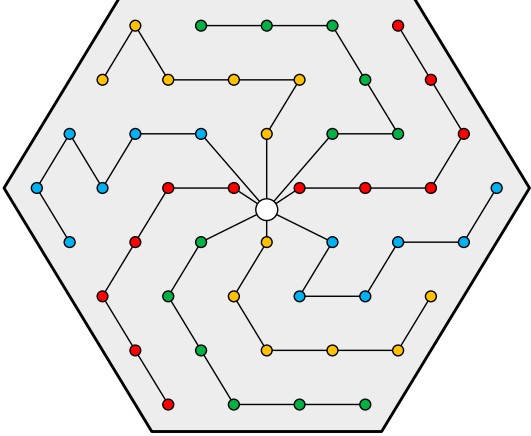

**Figure 4.** Brædstrup borehole thermal energy storage (BTES) system layout and serial BHE connections [33].

An important prerequisite for an accurate simulation of a real BTES system is good knowledge of the underground's thermal properties at the storage location. Tordrup et al. [31] used inverse modeling to determine thermal conductivity and volumetric heat capacity values for six geological layers at the storage site for a suitable model parametrization. Using monitoring data of the first 500 days of operation, they could reach an overall deviation between the energy balance of their fitted model and the monitoring data of 4.0%. The resulting average effective values for the thermal conductivity and the volumetric heat capacity of 1.72 W/(m K) and 1.96 MJ/($m^3$ K), respectively, were used for the parametrization of the *MoBTES* model. However, this study has some important limitations which have to be kept in mind for the interpretation of the case study results. For example, adiabatic boundary conditions were set at all model boundaries to reduce the computational effort of the inverse modeling process. This is a major limitation, especially for the ground surface boundary, where thermal losses of the storage are usually highest.

In contrast to Tordrup et al. [31], who used data smoothed to daily values, this study exploits the full temporal resolution of 5 min from the raw monitoring data. Although the available data covers the time from the initial storage startup until the end of 2017, Tordrup et al.'s parameter estimation only utilized data of the first 500 days of operation. Therefore, the performance evaluation of the *MoBTES* model is primarily conducted for this period as well. Nevertheless, selected *MoBTES* models are simulated over the whole range of available data and compared to the monitored energy balance.

Figure 5 shows the measured inlet and outlet temperatures of the storage system during the considered time span. It indicates that during this time span, different operation strategies were tested: In 2012, the first year of operation, charging and discharging was performed in pulses with constant volume flow rates. In contrast to that, during the second charging period in 2013, the outlet temperatures were kept constant at defined temperature levels over longer periods, presumably by regulation of the volume flow. The presence of different operational strategies and the comparably good availability of data make the Brædstrup dataset a particularly suitable test case for the validation of *MoBTES*.

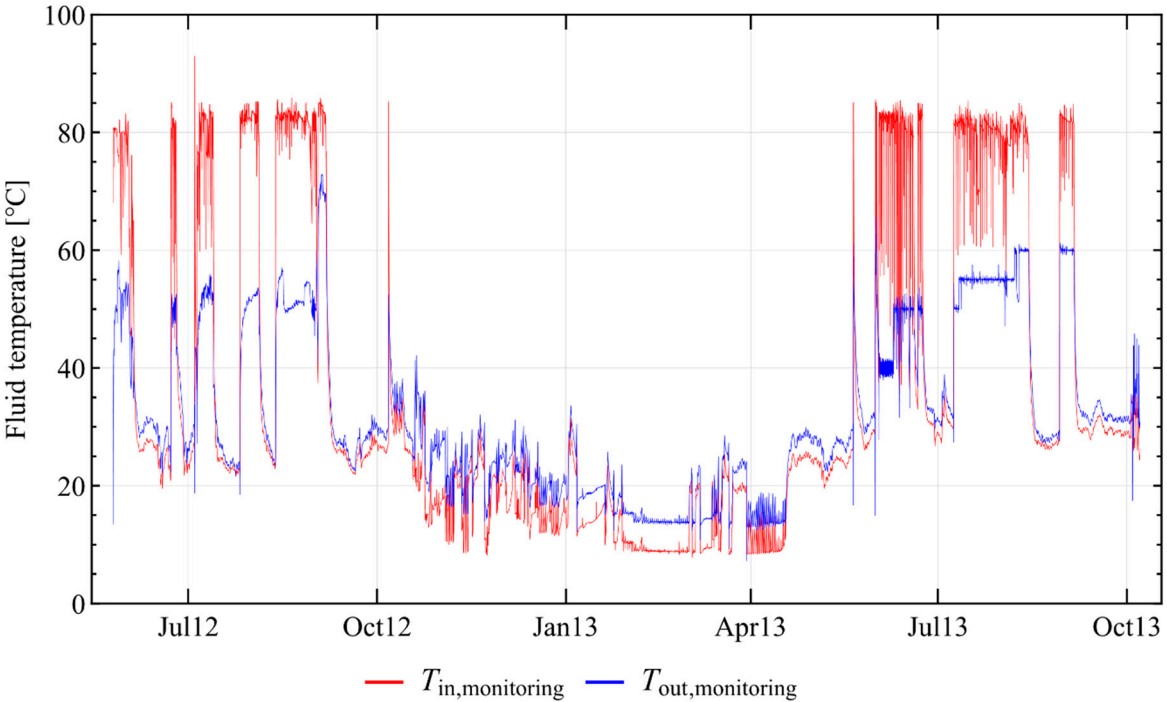

**Figure 5.** Measured inlet and outlet temperatures of the BTES system in Brædstrup during the first 500 days of operation.

For the comparison of the measurement data to the *MoBTES* models, all implemented modeling approaches are tested, using the monitored inlet temperature and volume flow rate time series as input. TRM and TRCM models are deployed for the BHE models and for the local model the steady flux and the FDM variants are compared. The FDM model's number of capacity nodes is varied between 2 and 16. Overall, a total of 20 simulations are carried out over the first 500 days of operation. Subsequently, the deviations between simulated and monitored values of the storage outlet temperature as well as the charged and discharged amount of energy are calculated for all models.

## 3. Results

### 3.1. Parameter Study Results

Different BTES layouts and geometries are simulated using *MoBTES* with both the FDM as well as the steady flux local model. Subsequently, the amounts of charged and discharged heat are calculated

and compared to the results of the outcome of the respective *FEFLOW* benchmark models. Figure 6 shows the individual deviations of charged and discharged energy of all simulated *MoBTES* models. For both model variants, regression lines are depicted for the deviation in charging and discharging to illustrate the impact of the storage volume.

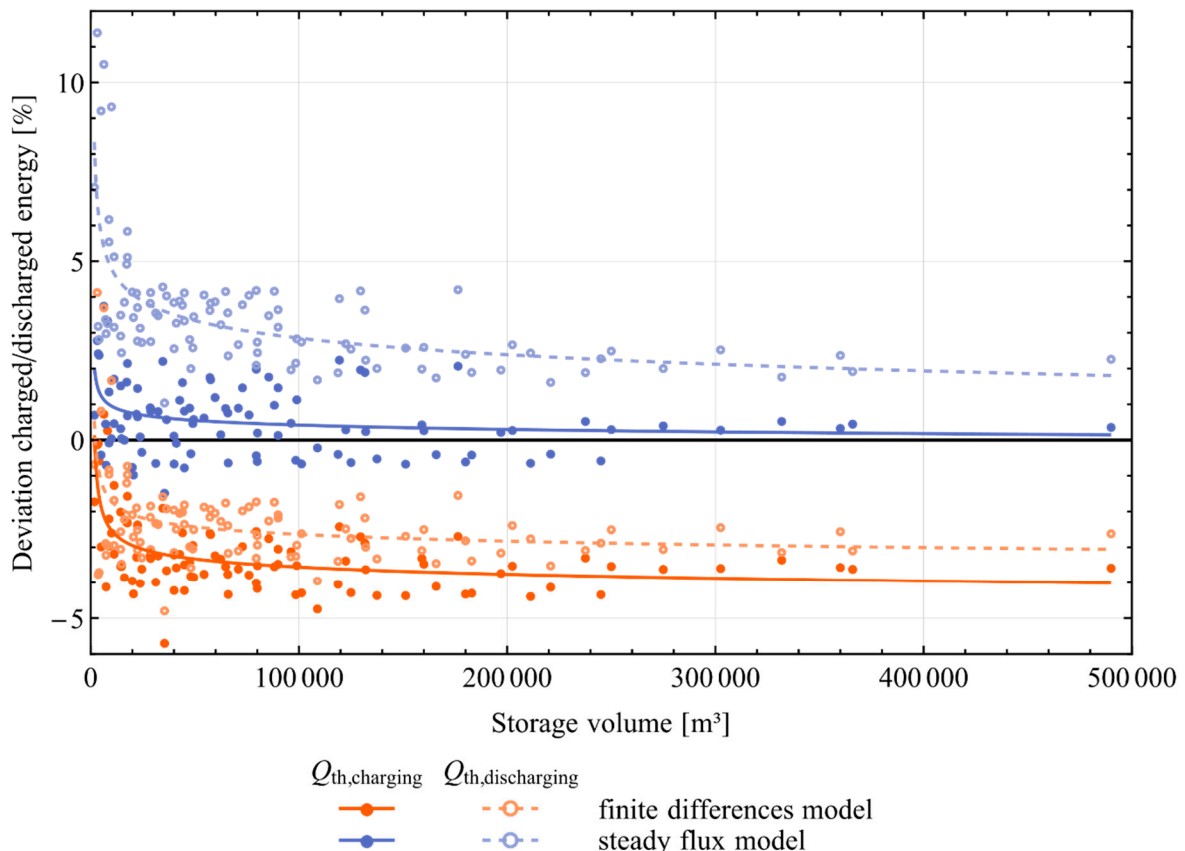

**Figure 6.** Relative deviation of charged and discharged thermal energy between *MoBTES* and *FEFLOW* for different storage system layouts.

Table 2 summarizes the mean results of the two local model variants. In comparison with the *FEFLOW* models, the FDM models underestimate the amount of charged energy by an average of −3.2% and discharged energy by an average of −2.3%. The similar magnitudes of deviation for both charging and discharging result in storage efficiencies close to those of the benchmark models. While the mean storage efficiency of all *FEFLOW* models is 60.9% the FDM *MoBTES* models yield an efficiency of 61.4%. In contrast to that, the deviations for charging and discharging differ significantly more for the steady flux models which achieve 62.5% storage efficiency on average. Underlying reason for this is a small deviation of +0.6% for charging in combination with an overestimation of +3.5% for the discharged energy amounts.

**Table 2.** Mean values of the *MoBTES* simulation results for both local model variants and deviation to *FEFLOW* benchmark models (standard deviation in brackets).

| Results | *MoBTES* FDM | *MoBTES* Steady Flux Model |
|---|---|---|
| Mean storage efficiency *MoBTES* | 61.4% | 62.5% |
| Mean deviation from *FEFLOW*: charged energy | −3.2% (±1.1%) | +0.6% (±1.0%) |
| Mean deviation from *FEFLOW*: discharged energy | −2.3% (±1.3%) | +3.5% (±1.7%) |
| Average computation time *MoBTES* | 751.8 s | 181.1 s |

On average, running an FDM model took 752 s, whereas the steady flux-based models required less than one-quarter of this time (181 s). Most of the computation time was consumed for the preprocessing, translation, and compilation of the *Modelica* models to C-code, whereas the actual simulations required only a fraction of it. The study was carried out using *SimulationX*, but for comparison, selected models were simulated in *Dymola* [34] and *OpenModelica* [35] as well. While *OpenModelica* yielded similar computation times as *SimulationX*, the model translation and compilation in *Dymola* took significantly less time resulting in much shorter overall computation times.

Figure 7 shows the impact of different parameters on the relative deviation between the charged and discharged energy amounts of *FEFLOW* and *MoBTES* FDM models. In accordance with Figure 6, the mean deviation in charging is larger than the deviation in discharging for all models utilizing an FDM local model. Additionally, it can be observed that the choice of layout seems to have an impact on the deviation, since there is a notable difference for circular, rectangular, and hexagonal layouts. Moreover, the deviation increases for more shallow systems and for larger BHE spacings.

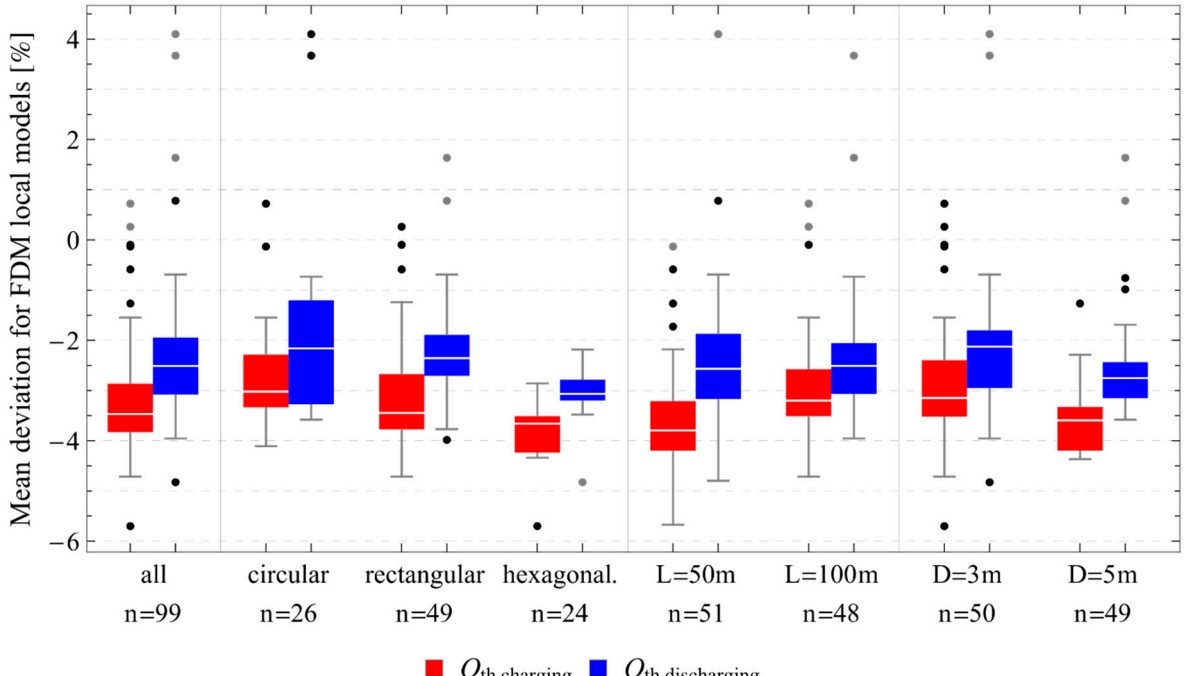

**Figure 7.** Impact of different parameters on the deviation of charged and discharged energy between *FEFLOW* and *MoBTES* models using an FDM local model (L = BHE length, D = minimal BHE distance).

Figure 8 illustrates the distribution of the deviations between *FEFLOW* and steady flux *MoBTES* models. During charging, the *MoBTES* models produce results very similar to *FEFLOW* and exhibit only few outliers. In contrast, the deviation is much larger during discharging and more outliers occur. The increased difference between relative deviations for charging and discharging leads to the higher discrepancy in storage efficiencies (cf. Table 2).

### 3.2. Case Study Results

For the case study, 20 *MoBTES* models with different modeling approaches and levels of discretization are simulated using monitoring data from the Bædstrup BTES inlet temperature as input. Subsequently, their outputs are compared to the outputs of the on-site measurements. One simulation of the considered 500-day-spanning monitoring data takes on average 12.3 min with models using the FDM approach for the local model. Compared to that, the steady flux approach reduces the computation time by more than 70% to only 3.5 min.

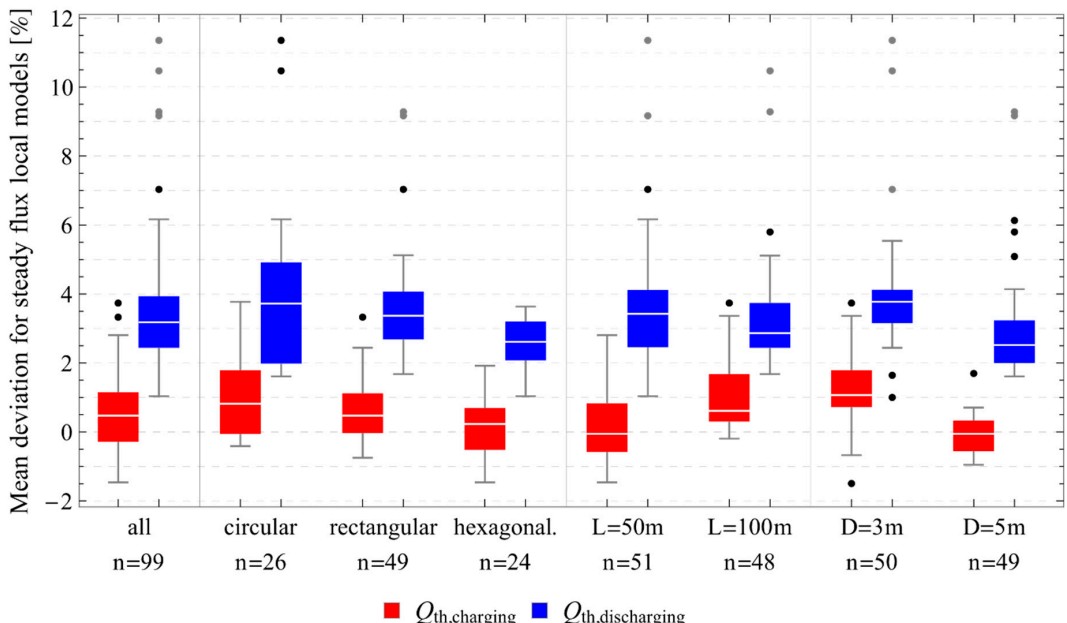

**Figure 8.** Impact of different parameters on the deviation of charged and discharged energy between *FEFLOW* and *MoBTES* models using a steady flux local model (L = BHE length, D = minimal BHE distance).

Figure 9 plots the computation times against the average temperature deviations for all individual simulation runs. In case of the FDM-based models, the computation time correlates with the level of discretization, i.e., the number of capacity nodes used. The steady flux models, which have only one capacity included, achieve a comparable computational speed to the FDM models with two capacities. This indicates that not only the absolute amount of DoFs determines the computational effort. Accordingly, model runs which utilize a TRCM BHE model do not generally take longer than their TRM counterparts, even though they have additional DoFs.

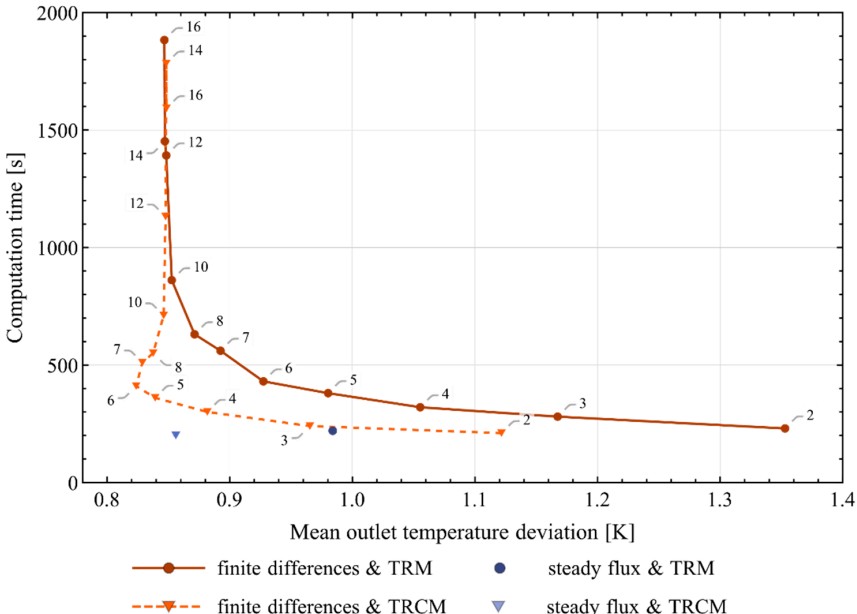

**Figure 9.** Computation time and mean deviation of the outlet temperature in comparison to the Brædstrup monitoring data for different local models. FDM models are labeled with their according number of used capacity nodes.

As a measure for the model accuracy, the mean outlet temperature deviation $\Delta T_{mean}$ is determined according to Equation (11), where $t_{sim}$ is the final simulation time and $\mathbb{1}(t)$ is an indicator function being 1 for times of storage operation and 0 otherwise:

$$\Delta T_{mean} = \frac{\int_0^{t_{sim}} \mathbb{1}(t) \left| T(t)_{outlet,monitoring} - T(t)_{outlet,MoBTES} \right| dt}{\int_0^{t_{sim}} \mathbb{1}(t) dt} \tag{11}$$

Regarding the outlet temperature deviation, models using a TRCM generally perform better than their counterparts with a TRM. Only for finer capacity node meshes both approaches' accuracies converge. Notably, a minimum in the deviation of the outlet temperature can be observed for the combination of TRCM models and FDM local models with six capacity nodes. A finer discretization does not improve the model's accuracy any further.

Figure 10 shows the deviation of the energy balance of the *MoBTES* models and the monitoring data. All models exhibit an underestimation of the energy balance by −3.2 to −8.4%. The lowest deviations can be observed for the models, which use a steady flux local model, showing an underestimation of the monitored energy balance of −3.2% for the model using a TRCM BHE model and −3.9% for the TRM variant, respectively. For the simulations utilizing FDM local models, a general trend towards lower deviations can be observed for an increase in the number of capacity nodes. Variants with six capacities or more exhibit an underestimation of the energy balance by approximately 6%. Regarding the BHE models, it can again be observed that models, which use the TRCM approach, show better accuracy in comparison to their TRM counterparts.

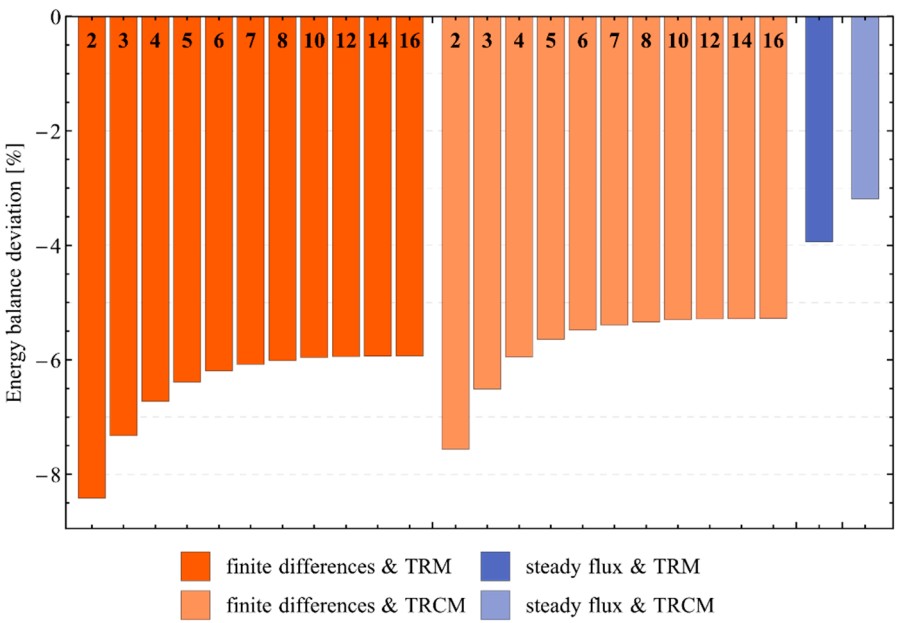

**Figure 10.** Relative deviation of the *MoBTES* models' energy balance in relation to the monitoring data after 500 simulated days (number of local model capacities on finite differences bars).

A more detailed insight into the model's short-time behavior can be gained by comparing the results of the different TRCM-based models to the monitored data over the first 24 h of BTES system operation (cf. Figure 11). At the initial startup of the system, the inlet temperature was kept relatively constant at a temperature of 80 °C, whereas the volume flow rate was adjusted over time. During the first twelve hours, models utilizing the FDM approach with two to five capacities or the steady flux model result in outlet temperatures above the monitored outlet temperature, whereas FDMs with seven or more capacities underestimate it. The model, which uses an FDM with six capacities, shows an almost perfect fit. It can be observed that the outlet temperature of the FDM variants converges to a

certain profile for an increasing number of capacity nodes. For the second half of the shown period, all models overestimate the outlet temperature, exhibiting smaller deviations for models with a finer discretization of the capacity.

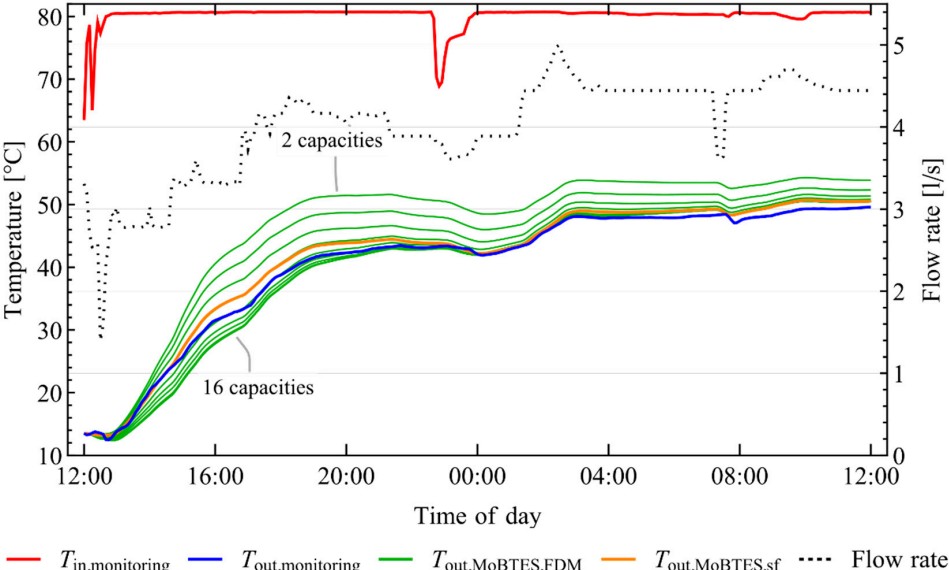

**Figure 11.** Comparison of monitored and simulated outlet temperatures for *MoBTES* models with Thermal Resistance and Capacity Model (TRCM) (FDM variants: number of capacity nodes increases from top to bottom).

To test the predictive abilities of *MoBTES* and the underlying parametrization of the case study over a larger time span, a steady flux model and an FDM model with six capacities (both using the TRCM BHE model) are simulated over the whole range of available monitoring data of 1680 days. Subsequently, the energy balance histories of the models are calculated and compared to the monitored energy balance history (Figure 12).

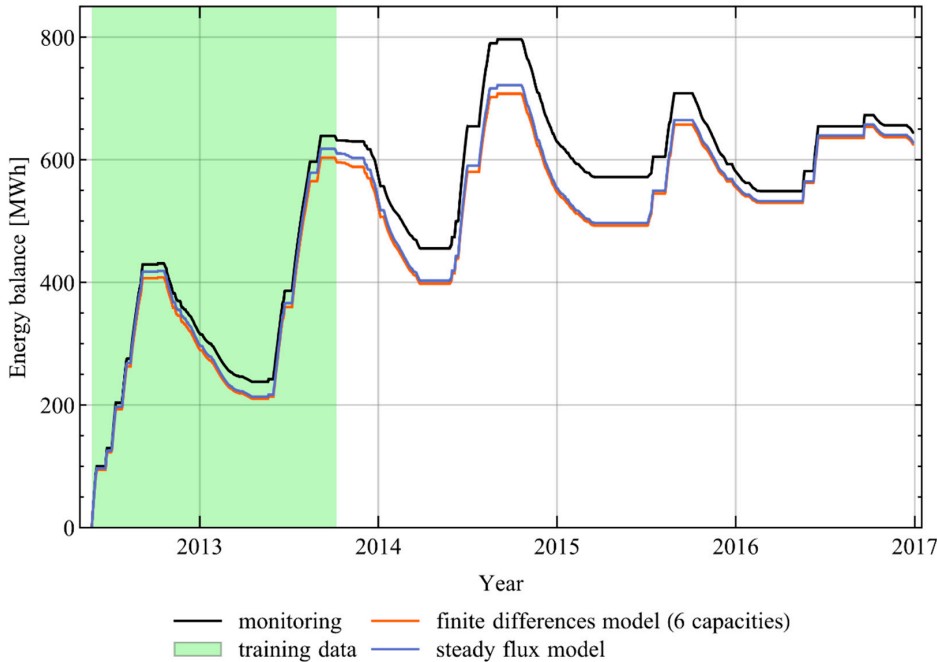

**Figure 12.** Energy balance from monitoring data and two selected models from startup until the end of the available data.

The final energy balance deviation after 1680 days amounts to −2.9% and −2.4% for the FDM approach and the steady flux approach, respectively. Surprisingly, these values are considerably lower than the deviations observed after 500 days of simulation (−5.5% and −3.2%). More important though is the maximum deviation which occurs during the summer of 2015 and amounts to −13.9% and −13.1%, respectively. The difference between the energy balances of the two *MoBTES* model variants is small in comparison to their deviation to the monitored data.

## 4. Discussion

### 4.1. Parameter Study

The parameter study's main purpose is to compare *MoBTES* to an established model with well-defined parameters and thereby assess its ability to accurately predict the amount of charged and discharged thermal energy over a large parameter range. However, the *FEFLOW* models, which were chosen as benchmarks, also represent a simplification of realty and are prone to the effect of geological uncertainty. Consequently, *FEFLOW*'s actual accuracy in terms of simulating real BTES systems is limited. The comparison of *FEFLOW* and *MoBTES* should therefore be regarded as validation of the new model by numerical means.

The results presented in Section 3.1 show minor deviations between the charged and discharged amount of heat for all *MoBTES* models, except for very small BTES systems (cf. Figure 6). However, BTES systems consisting only of very few BHEs are not common, as storage efficiency generally increases with size, making those small systems inefficient. Therefore, the inaccuracy for small systems does not represent a major limitation of *MoBTES*. It is most likely caused by assumptions made for the modeling approaches of *MoBTES* local models. Figure 1 illustrates that the local models exchange thermal energy with the BHEs through the borehole walls and with the global model by extracting and injecting heat to their whole volume. As a consequence, no heat is exchanged via their outer boundaries. This poses a reasonable assumption for BHEs in the center of BTES systems. However, the error is larger for BHEs at the edge of the storage volume as they are not symmetrically surrounded by neighboring BHEs. The surface-to-volume ratio increases for small systems, hence leading to larger deviations. As a consequence, the minimal possible number of BHEs in *MoBTES* was limited to seven, to avoid excessive errors.

The overall magnitude of deviation for most models is in a low single-digit percentage range, which can be regarded as adequate. Accordingly, it is not possible to identify a local model approach as more accurate for simple step-response studies like the presented parameter study. However, additional information can be drawn from it. Although, the steady flux local models exhibit comparable magnitudes of deviation as the FDM variants, they clearly outperform them regarding the computational efficiency (cf. Table 2). Consequently, the steady flux model is preferable for large parameter studies with steady operation schemes.

Interestingly, the choice of the BTES layout seems to have an impact on the deviation for both local model variants (cf. Figures 7 and 8). Circular layouts tend to higher estimates in comparison to rectangular layouts, hexagonal resulting in the lowest estimates. There are several possible underlying reasons, which have to be discussed:

The first one concerns the overall shape of the BTES systems and the global models. We have to remember that *MoBTES* itself is a circular BTES model, and therefore the actual shapes of the benchmark models differ for rectangular and hexagonal layouts (cf. Figure 3). If this geometry transformation was the root cause for the observed layout impact, the circular models would have to perform best. This is true for the FDM variants, where the magnitude of deviation is smallest for circular arrangements, but not true for the steady flux variants, where the opposite is the case. Consequently, the model reduction approach of the global model has probably only a minor effect.

A second potential cause regards the shape of the local models. These have a circular cross section, whereas none of the areas around a single BHE is circular for the actual layouts. However, this

deviation in shape is considerably smaller for hexagonal cross-sections in comparison to rectangular ones. Again, the fact that hexagonal layouts perform best for the steady flux variants, while they exhibit the largest deviations for FDM variants contradicts the assumption that this might be the major cause for the impact of the choice of layout on the deviation.

A third potential cause for the observed impact of the storage layout on the deviation could be their diverging packing densities. Equal minimal BHE distances D lead to different radii of the local model $r_{loc}$ for rectangular, hexagonal, and circular layouts. Hexagonal layouts yield the highest packing density, resulting in smaller radii $r_{loc}$ for the same BHE distances D. Therefore, parameter study models with equal minimal BHE distance D, but varying storage layouts, result in different radii $r_{loc}$. If this would be the underlying reason, the observed impact would actually have to be related to the model volume, which is directly correlated to the BHE distance D and length L. Hexagonal layouts, which have the highest packing density, i.e., smaller storage volumes in average, yield in lower estimates of charged and discharged amounts of energy for both local model variants. However, the impact of BHE distance D and length L (cf. Figures 7 and 8) suggests that a potential correlation between deviation and storage volume should be reversed.

All effects, which are implied by the presented figures, are rather small and possibly not statistically significant. Nevertheless, some useful conclusions can be drawn by analyzing the difference in deviation between charging and discharging for the steady flux models (cf. Figure 8 There is a higher number of more pronounced outliers for discharging than for charging. Referring to Figure 6, these outliers can be connected to models of small BTES systems. This indicates that steady flux models with very small storage volumes exhibit the highest deviations from the *FEFLOW* benchmarks of all simulated *MoBTES* models. Therefore, FDM local models should be preferably applied for the simulation of small BTES systems.

As an overall result of the parameter study, it can be stated that *MoBTES* can be used for the simulation of simple BTES applications, as it produces results very close to those of detailed FEM models. Still, this is limited to certain applications. For example, *MoBTES* does not consider groundwater flow or BHE arrangements that strongly diverge from axisymmetric layouts. However, BTES systems are preferably built on sites with negligible groundwater flow to reduce convective losses. In addition to that, these systems should be constructed with a low surface-to-volume ratio to ensure an efficient performance. Due to these general rules for the construction of BTES systems, the aforementioned limitations do not pose a problem for most practical purposes. These general rules do not necessarily apply for regular BHE arrays, which are only used either for extraction or for injection of heat. This underlines the importance of using *MoBTES* for its original purpose of storage applications.

### 4.2. Case Study

Design and operation of the Brædstrup system were investigated in great detail to achieve an accurate representation by the MoBTES models [31,33]. Nevertheless, there are still significant uncertainties left regarding the components' thermal properties. This has to be kept in mind in order to avoid overinterpretation when comparing the simulation results to the monitoring data. Independently, the case study is perfectly suited for a further comparison of the different *MoBTES* model variants in terms of efficiency and accuracy.

#### 4.2.1. Computational Effort and Mean Outlet Temperature Deviation

While the simulation of BTES systems using *MoBTES* reduces the computational effort in comparison to 3D FEM models by several orders of magnitude, there are significant differences between the different *MoBTES* approaches as well: computation times for the initial 500-day period range from 200 s to 1880 s (Figure 9). As expected, the specific computation time for the simulation of one year of storage operation is considerably higher compared to the parameter study described in Section 3.1. For example, the combination of steady flux local model and TRM BHE model takes 165.1 s per simulated year for the case study. In contrast, a comparable *MoBTES* model from the parameter

study takes only 20.6 s for one year. This can probably be attributed to generally smaller time steps in the case study due to a more transient operation, on the one hand, as well as a limitation of the maximum time step size to the resolution of the monitoring data of 5 min on the other hand.

Putting the computational time and the mean outlet temperature deviation of the different models in relation to each other (Figure 9) also unveils some interesting coherencies. As expected, the computational effort generally increases with the number of capacities of the local models. Consequently, the steady flux local models, which include only one capacity reach the lowest computation times. Comparing the temperature deviation of TRM-based and TRCM-based combinations strongly emphasizes the superiority of the TRCM approach. Both the FDM-TRM as well as the FDM-TRCM model combinations' mean outlet temperature deviations decrease with an increase in the number of capacities. However, for higher capacity numbers they seem to converge to a certain minimal value, which is a common outcome for grid refinement studies. However, for all models comprising less than ten capacities, the TRCM approach achieves in some cases significantly lower deviations than the comparable TRM models, while resulting in slightly lower computation times. As stated several times, the consideration of the grout's thermal capacity is of high importance. Discretization of meshes for numerical simulation should generally be refined at model areas with steep gradients and strong transient behavior. The temperatures within the boreholes of a BTES system fulfill both aspects, which explains the good performance of the *MoBTES* variants that use TRCM BHE models.

It seems that there is a good trade-off between the computational effort and the accuracy for models with six to eight capacities: these models gain a significant reduction in the temperature deviation while they experience only a minor increase in the computational time. Surprisingly, the deviation for FDM-TRCM model combinations even exhibits a minimum for six capacities, which might be explained by a closer look at the short time performance of the models (see Section 4.2.3). In terms of a good trade-off between computational speed and accuracy, the steady flux model combined with a TRCM model seems to be a serious alternative to the FDM based models.

### 4.2.2. Comparison of Overall Energy Balance Deviations

Counterintuitively, the overall energy balance deviations (cf. Figure 10) of the models utilizing the steady flux approach even undercut that of the FDM based models. However, this finding does not contradict the observation that some of the FDM models perform better in terms of predicting the outlet temperature (cf. Figure 9). This is because of two reasons: First, the underlying energy balance is achieved by the summation of charged and discharged energy. Consequently, an error in charging can be compensated by an equally large error in discharging. In contrast to that, the temperature deviation considers only absolute deviation values, as defined by Equation (11). Second, the temperature deviation does not take the volume flow rate into account. Since the volume flow rate varies significantly over time, the simulated outlet temperature contributes to the energy balance with varying weights.

### 4.2.3. Short Time Accuracy

The presented results are values integrated over the entire simulated time span. Therefore, they are mostly defined by the long-term accuracy of the model, whereas the short-time accuracy can be investigated best at times of strong changes in the operation of the BTES. The start-up phase of the storage represents such a sudden change in operation, which can be regarded as a step-pulse with a temperature raise of almost 70 K (cf. Figure 11). Moreover, the strong variation of the volume flow rate during that time span poses an additional difficulty for the reproduction of the transient operation behavior of the system. Most likely, the model with six capacities represents the best fit to the monitored outlet temperature by coincidence. However, this could explain the minimum in the mean outlet temperature deviation for this number of capacities (cf. Figure 9). In line with the results for the average outlet temperature deviation (cf. Figure 9), the steady flux model performs equally

well as the FDM model with five capacities. As expected, the difference between the model variants decreases over time.

### 4.2.4. Comparison of Model Results and the Extended Monitoring Data

A successful comparison of the *MoBTES* models to the monitored data requires a good knowledge of the subsurface properties. For the Brædstrup case study, these key input parameters are gathered from a parameter estimation study conducted by Tordrup et al. [31]. With this parametrization, *MoBTES* underestimates the amount of energy after 500 days of operation by 3.2% to 6.5% disregarding models that use local FDM models with four or less capacities (cf. Figure 10). For comparison, the 3D FEM model by Tordrup et al. [31] with the best fit also resulted in an underestimation of the energy budget by 4%. This indicates that *MoBTES* is capable to reproduce the operation of the Brædstrup system during this period just as well as the 3D FEM model.

Regarding the energy balance history for the extended simulation period of 1680 days (cf. Figure 12) a maximum deviation of the *MoBTES* energy balance can be observed in 2015 before a trend reversal sets in resulting in a comparably low deviation by the end of the simulation period. The change in the model drift could indicate an insufficient size of the modeled region. However, this concern could be dispelled as a significant increase of the outer model boundaries did not result in any noticeable change in the models' energy balance.

When assessing the predictive capabilities of *MoBTES*, the limitations of the inverse modeling study by Tordrup et al. mentioned in Section 2.2.2 should be considered. Especially, the application of the adiabatic model boundaries represents an oversimplification as it corresponds to a perfect insulation on the ground surface. This is of little consequence during the initial storage operation when temperatures inside the storage volume and consequently the losses through the ground surface are naturally low. Therefore, this could explain both the good fit during the start-up period and the following increasing deviation.

The decline in the deviation towards the end of the simulation is an indication that the meaningfulness of an overall balance is limited, as different segments of the energy balance compensate each other. However, the use of the overall energy balance as evaluation value is due to the lack of more detailed data of the original simulations of Tordrup et al.

If an overall evaluation of the model quality is to be given, it must be considered that model input parameters, such as heat capacities and thermal conductivities of the soil and grout are subject to considerable uncertainties. Under this premise, the results of the FDM models with at least five capacities and the results of the steady-flux model are sufficiently accurate. As all models have some difficulties to accurately predict the storage behavior beyond the fitting period, the parameter estimation study should be repeated on the data foundation that is available now, taking thermal losses through the ground surface into account. As this goes clearly beyond the scope of this study, it should be considered as a future application of *MoBTES*.

### 5. Conclusions

The presented *MoBTES* model facilitates the deployment of different modeling approaches for its sub-models, allowing for an adaption to the numerical requirements of varying applications. The currently implemented variants are based on well-known and proven approaches, which exploit the physical characteristics of BTES systems. Consequently, the comparison of *MoBTES* to 3D FEM benchmark models and monitoring data from an existing plant reveals only minor deviations in their performance figures. While all variants are able to adequately reproduce the long-term system behavior, the right choice can significantly increase the computational efficiency or short-time accuracy. In addition to that, the presented model framework can be used as a test bed for new developed modeling approaches, provided they are compatible with the division of the model into a global, a local and a BHE sub-model. Other possible future applications of the developed open source *Modelica* library, could

be the rededication of the BHE models for non-storage applications or the realization of additional underground thermal energy storage technologies by reusing the available ground components.

In contrast to existing BHE models in *Modelica*, *MoBTES* is a dedicated BTES model and therefore should cover all relevant design features of such systems. Therefore, emphasis was put on the implementation of functionalities like serially connected BHEs, consideration of the stratigraphy at the storage site, flow reversal, hydraulic pressure loss, or partly insulated BHEs. To ensure an efficient operation, actual BTES systems are favorably build as compact arrays of thermally interacting BHEs on sites with negligible groundwater movement. Therefore, *MoBTES* only considers those cases and its accuracy might be impaired significantly for other applications.

The implemented model features allow for an accurate assessment of the impact of different designs on the storage performance, while maintaining a computational efficiency suitable for system simulation. Additionally, the flexibility of *MoBTES* enables the use of very fast models for extensive parameter studies or stochastic simulation. The versatile and multi-domain modeling approach of *Modelica*, allows for the integration of *MoBTES* into models of whole energy systems, including sector coupling and the combination with a wide range of other open source model libraries.

**Supplementary Materials:** The following are available online at http://www.mdpi.com/1996-1073/13/9/2327/s1, Table S1: Benchmark parameter study models and results. The MoBTES Modelica library, including an example package with the parameter study model, the case study model and all related parameters or datasets, is available online at https://github.com/JFormhals/MoBTES.

**Author Contributions:** This research was made collaboratively by all authors. Conceptualization was done by J.F., B.W., D.O.S., and I.S. Development of the methodology, software, data processing, formal analysis, and visualization was done by J.F. Validation was done by H.H., B.W., and J.F. Funding acquisition and supervision was done by D.O.S. and I.S. Review and editing was done by D.O.S., H.H., B.W., I.S., and J.F. All authors have read and agree to the published version of the manuscript.

**Funding:** This work was financially supported by the German Research Foundation (DFG) in the framework of the Excellence Initiative, Darmstadt Graduate School of Excellence Energy Science and Engineering (GSC 1070) and the European Research Development Fund (ERDF) by supporting the North-West Europe Interreg project DGE-Rollout.

**Acknowledgments:** The authors want to thank Brædstrup Fjernvaerme for sharing the monitoring data of the solar district heating plant in Brædstrup that was used for this work. Furthermore, the authors want to thank Morten Vang Bobach of Arcon-Sunmark and Jim Larsen of Brædstrup Fjernvaerme for sharing their detailed knowledge about the construction and operation of the BTES system.

**Conflicts of Interest:** The authors declare no conflicts of interest.

## Nomenclature

| | |
|---|---|
| BHE | borehole heat exchanger |
| BTES | borehole thermal energy storage |
| DoF | degrees of freedom |
| FDM | finite differences model |
| HSRM | hybrid step response model |
| MoBTES | Modelica borehole thermal energy storage model |
| SBM | superposition borehole model |
| TRCM | thermal resistance and capacity model |
| TRM | thermal resistance model |

## Symbols

| | | |
|---|---|---|
| a | thermal diffusivity | $m^2/s$ |
| c | gravimetric thermal capacity | $kg/m^3$ |
| C | thermal capacity | J/K |
| D | borehole spacing | m |
| L | Borehole length | m |
| q | specific heat flux | W/m |
| Q | Thermal energy | J |
| $\dot{Q}$ | heat flux | W |
| r | radius | m |
| R | thermal resistance | K/W |
| T | temperature | K |
| t | time | s |
| δ | relative deviation | - |
| ρ | density | $kg/m^3$ |
| λ | thermal conductivity | W/(m K) |
| τ | time constant | s |
| 𝟙 | indicator function | - |

## Subscripts

| | |
|---|---|
| b | borehole wall |
| glo | global problem |
| loc | local problem |
| m | mean |
| min | minimum |
| sf | steady flux |
| sim | final simulation time |
| th | thermal |
| 0 | constant temperature profile under steady flux condition |

## Appendix A  Modelica Library

The structure of the developed *Modelica* library can be seen in Figure A1. The main component is the *BTES* model, which has one fluid port each for inlet and outlet of the storage. An additional input is available if the user choses to define a time-varying ground surface temperature. All physical components, which are used to build the BTES model, are included in in the *Components* package. The *Components.Ground* package includes models for the global solution and the different local solutions, whereas the *Components.BoreholeHeatExchangers* package includes the Single-U, Double-U and Coaxial BHE models. The *Builder* package includes functions and enumerations which are needed for the assembly of the *BTES* model. All parameter sets which can be used in the BTES model are stored in the *Parameters* package as records. There are typical data records and the records of the examples shown in this work for the location, different soil types, the heat exchangers, grouts and heat carrier fluids. All components or parameter records that can be replaced by each other share common base classes. These base classes define all common properties, like interfaces or indispensable parameters, which is especially important for the replacement of the local, global, and BHE models. New implementations should inherit from the respective base class, to be in conformity with the *MoBTES* modeling approach. *MoBTES* version 1.0 has been successfully tested for *SimulationX* and *Dymola*.

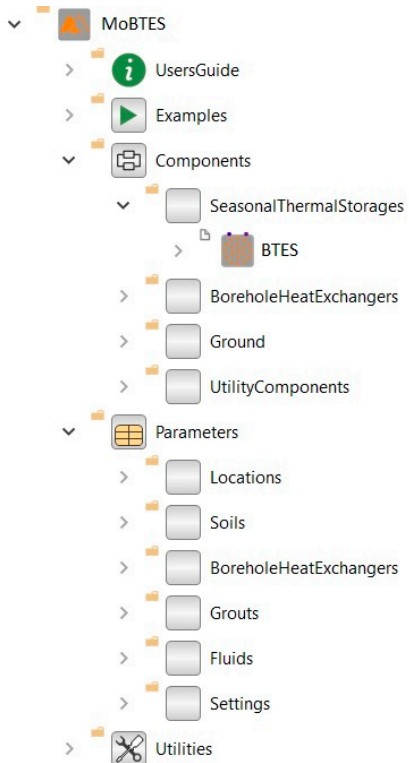

**Figure A1.** *MoBTES Modelica* library structure.

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
