# Peer review of "A Modelica Toolbox for the Simulation of Borehole Thermal Energy Storage Systems"

_energies, doi:10.3390/en13092327_

Round 1

Reviewer 1 Report

Borehole thermal energy storage it is an interesting topic take into consideration the energy storage problematics. 

The strength of this paper is given by the quality of presentation.

The case study it is an interesting one, especially because of the comparation between model results and monitoring data.

Author Response

Dear reviewer,

thank you very much for the time and effort you have put into the review of our manuscript. This is appreciated very much.

You did not have any specific points, which we could adress. The main changes in the revised manuscript regard the length. The content was kept the same, but, wherever possible, information that was not necessary for the understanding of the work, was deleted. The overall length of the text has decrease by ~10%.

Best regards

Reviewer 2 Report

The authors develop and test a simulation module for the study Borehole Thermal Energy Storage Systems. The numerical model is validated by comparison with a 3D finite element method (FEM) and with existing data. The authors present a detailed and very complete work.

To improve the paper the authors should address the following minor issues:

  1. Improve the description of figure 1. It should be related to the actual physical situation and each component of the figure should be identified and describe in the caption.
  2. The text of the paper is too long. Readability would improve if the authors could reduce text length focusing on important issues and reducing repetition. Note that the number of figures is adequate and all figures should be kept. 

Author Response

Dear reviewer,

thank you very much for the time and effort you have put into the review of our manuscript. This is appreciated very much. You provided two possible ways to us, which would improve the quality of the manuscript. Since all reviewers agreed on the general relevance of the investigations, we tried to adress your points without changing the content.

Point 1: “Improve the description of figure 1. It should be related to the actual physical situation and each component of the figure should be identified and describe in the caption.”

Answer: The four components shown in the figure have been addressed in the caption and are now explained briefly. Further elaborations at this position would exceed the common limitations of the length of captions. A deeper discussion can be found in the text. We are not completely sure how the remark about the actual physical situation is meant, but we will address this point to our best: Figure 1 is the most important figure of the manuscript for the understanding of the fundamental model structure and the capability to use different modeling approaches. We have found, that especially for readers not familiar with this division into sub-models, it can be hard to grasp. Therefore, Figure 1 was intentionally kept on a low level of detail, keeping the focus on the structure and highlighting the possibility to fill in the void space inside the sub-models with arbitrary modelling approaches.

Point 2: “The text of the paper is too long. Readability would improve if the authors could reduce text length focusing on important issues and reducing repetition. Note that the number of figures is adequate and all figures should be kept.”

Answer: The reviewer states a valid point. Especially the case study with real monitoring data required more text than anticipated, since there are a lot of points which have to be considered. We tried to reduce the length of the text by erasing repetitions and keeping a slightly stronger focus, without leaving out any relevant information from the original manuscript. The total number of words was reduced from 9,355 to 8,580. We hope this is sufficient.

Best regards

Reviewer 3 Report

The paper is very well written, however it can be very dense at times and takes time for the reader to digest it. It is of acceptable quality for direct publication in this journal.

Author Response

Dear reviewer,

thank you very much for the time and effort you have put into the review of our manuscript. This is appreciated very much.

You did not have any specific remarks, besides that the text is pretty dense at times. We agree with you and therefore tried to make it a little easier for the reader. We did not change the content, but did the following steps:

  • the text was read by people not familiar with the topic of thermal storage to identify sentences that are hard to understand
  • the identified sentences were rephrased to simplify them
  • a few complicated sentences, which are not necessary for the understanding of the study, were excluded
  • some explanations were relocated to other positions in the text
  • the overall length was reduced by ~10%

We are confident that these changes improve the quality of the manuscript.

Best regards